# Ruptured Posterior Inferior Cerebellar Artery Aneurysms: Integrating Microsurgical Expertise, Endovascular Challenges, and AI-Driven Risk Assessment

**DOI:** 10.3390/jcm14155374

**Published:** 2025-07-30

**Authors:** Matei Șerban, Corneliu Toader, Răzvan-Adrian Covache-Busuioc

**Affiliations:** 1Department of Neurosurgery “Carol Davila”, University of Medicine and Pharmacy, 050474 Bucharest, Romania; mateiserban@innbn.com (M.Ș.); razvancovache@innbn.com (R.-A.C.-B.); 2Department of Vascular Neurosurgery, National Institute of Neurology and Neurovascular Diseases, 077160 Bucharest, Romania; 3Puls Med Association, 051885 Bucharest, Romania

**Keywords:** PICA aneurysm, posterior circulation aneurysm, microsurgical clipping, endovascular treatment, far-lateral approach, cerebrovascular hemodynamics, artificial intelligence in neurosurgery, aneurysm genetics, flow diversion, computational fluid dynamics

## Abstract

**Background/Objectives:** Posterior inferior cerebellar artery (PICA) aneurysms are one of the most difficult cerebrovascular lesions to treat and account for 0.5–3% of all intracranial aneurysms. They have deep anatomical locations, broad-neck configurations, high perforator density, and a close association with the brainstem, which creates considerable technical challenges for either microsurgical or endovascular treatment. Despite its acceptance as the standard of care for most posterior circulation aneurysms, PICA aneurysms are often associated with flow diversion using a coil or flow diversion due to incomplete occlusions, parent vessel compromise and high rate of recurrence. This case aims to describe the utility of microsurgical clipping as a durable and definitive option demonstrating the value of tailored surgical planning, preservation of anatomy and ancillary technologies for protecting a genuine outcome in ruptured PICA aneurysms. **Methods:** A 66-year-old male was evaluated for an acute subarachnoid hemorrhage from a ruptured and broad-necked fusiform left PICA aneurysm at the vertebra–PICA junction. Endovascular therapy was not an option due to morphology and the center of the recurrence; therefore, a microsurgical approach was essential. A far-lateral craniotomy with a partial C1 laminectomy was carried out for proximal vascular control, with careful dissection of the perforating arteries and precise clip application for the complete exclusion of the aneurysm whilst preserving distal PICA flow. **Results:** Post-operative imaging demonstrated the complete obliteration of the aneurysm with unchanged cerebrovascular flow dynamics. The patient had progressive neurological recovery with no new cranial nerve deficits or ischemic complications. Long-term follow-up demonstrated stable aneurysm exclusion and full functional independence emphasizing the sustainability of microsurgical intervention in challenging PICA aneurysms. **Conclusions:** This case intends to highlight the current and evolving role of microsurgical practice for treating posterior circulation aneurysms, particularly at a time when endovascular alternatives are limited by anatomy and hemodynamics. Advances in artificial intelligence cerebral aneurysm rupture prediction, high-resolution vessel wall imaging, robotic-assisted microsurgery and new generation flow-modifying implants have the potential to revolutionize treatment paradigms by embedding precision medicine principles into aneurysm management. While the discipline of cerebrovascular surgery is expanding, it can be combined together with microsurgery, endovascular technologies and computational knowledge to ensure individualized, durable, and minimally invasive treatment options for high-risk PICA aneurysms.

## 1. Introduction

Intracranial aneurysms originating from the posterior inferior cerebellar artery (PICA) can be classified as one of the rarest subtypes of saccular aneurysms globally and represent the development of 0.5–3% of each intracranial aneurysm and approximately 10% of posterior circulation aneurysms [1]. The true incidence of PICA aneurysms remains difficult to ascertain because of the lower detection rates of unruptured PICA aneurysms and the high percentage of PICA aneurysms that are detected only after rupture, which also makes adequate detection difficult [2]. Ruptured PICA aneurysms have higher morbidity and mortality rates compared to ruptured anterior circulation aneurysms, with case fatality rates ranging from 20% to 35% in some instances [3]. PICA aneurysms are complicated by the unique anatomical relationship with the medulla oblongata, lower cranial nerves, and perforators of the brainstem. These three, when treated by traditional neurosurgical means, illustrate that neither surgical nor endovascular approaches can reliably prevent complications related to ruptured and unruptured PICA aneurysms [4].

PICA aneurysms are epidemiologically distinct from more common intracranial aneurysms. PICA aneurysms have been reported in some cases to be found slightly more often in men than women, and the average age at diagnosis is 50–60 years [1]. Hypertension, smoking, and connective tissue disorder have been implicated as risk factors in large multicenter studies [5]. With regard to location, geographical differences have been reported in PICA aneurysm detection. For instance, Asian and North American studies report significantly higher rates of detection of PICA aneurysms [6,7]. The etiology of these differences remains uncertain, but a possible explanation may be related to differences in the availability of diagnostic imaging, screening protocols, or genetic predisposition to form intracranial aneurysms in these groups [8]. Distinct from anterior circulation aneurysms, which are often detected in asymptomatic patients undergoing screening, the PICA aneurysm is often discovered after rupture and often presents with abrupt-onset headache, vertigo, dysphagia, and lower cranial nerve dysfunction related to the aneurysm location and similar to other posterior circ distance (PC) aneurysms [9].

Treatment options for PC aneurysms are still debated, especially considering the advantages of endovascular treatment of aneurysms. While microsurgical clipping is considered the gold standard for treating PC aneurysms, with a recent series having over ninety percent (90%) obliteration of the aneurysm and successful patient outcomes following direct surgical access (since inception), developments in neuroendovascular techniques and technology support initially treating many PC aneurysms endovascularly with fewer intraoperative morbidities [10]. Despite recent advances, PICA aneurysms are still very difficult to treat using endovascular techniques, considering the small diameter of the parent artery, distorted vertebrobasilar anatomy, and brainstem perforators that are frequent in this region of the brain. It has been reported in some studies that coiling PICA aneurysms endovascularly produced higher recurrence rates (up to 40% compared to fundus clipping), primarily from the risk of coil compaction, compromise of the parent vessel, and incomplete neck coverage [11]. Flow diversion stenting has changed the treatment of anterior circulation aneurysms; however, it is mostly unsuitable for the treatment of PICA aneurysms because of the risks of perforator infarction and the unpredictable hemodynamic effects it may have on PICA perfusion. Microsurgical clipping is the definitive treatment for PICA aneurysms when endovascular therapy is considered high risk or unsuitable [12]. The far-lateral approach affords the most direct path to aneurysms arising from the vertebrobasilar junction, where proximal control of the vertebral artery can be obtained and the neck of the aneurysm can be safely dissected with little cerebellar retraction [13].

The far-lateral microsurgical approach has evolved in recent years to include variants of the far-lateral approach, such as condylar resections and tailored neural opening plans with exposure that has reduced the risk of cranial nerve deficit [14]. Recent extensive retrospective reports have noted complete microsurgical obliteration rates for PICA aneurysms of 85–95%, with favorable long-term outcomes when performed at a cerebrovascular center with specific experience in treating PICA aneurysms [15,16].

While ultimately rare, the surgical management of ruptured PICA aneurysms remains relevant in the vascular neurosurgery literature, considering the risk of rebleeding, morbidity, and potential challenges/technical difficulties in achieving durable exclusion of these aneurysms [17]. This case outlines the successful microsurgical clipping of a ruptured, broad-necked PICA aneurysm with emphasis on the need for anatomical dissection, clip placement, which reflects the individualized surgical approach, and modifying surgery to develop good outcomes. When compared to other published cases of ruptured PICA aneurysms, this aneurysm presented unique challenges given its acute high angulation and deep positioning within the lateral medullary cistern. This required a modification of the dissection plane and a deliberate, tailored clip strategy to safely achieve adequate exclusion of the broad-necked aneurysm while maintaining flow-distal to the PICA. Further, the post-operative trajectory of the patient was uneventful with full neurological recovery and no delayed ischemic event, confirming the efficacy of having a peri-operative management framework.

Using this case, we hope to provide a thorough, evidence-based contribution to the PICA aneurysm literature, synthesizing modern microsurgical principles and current treatment paradigms in neurovascular surgery. We will systematically analyze the anatomical, technical, and clinical domains that contribute to surgical decision-making, ultimately refining best practices for a rare but high-risk lesion while contributing to the changing dynamic of managing aneurysms of the posterior circulation.

## 2. Case Presentation

A 66-year-old male with an extensive history of chronic ethanol abuse came to the Emergency Department with an acute onset of a thunderclap headache of maximum severity, followed by cyclical projectile vomiting and rapid progressive confusion. His symptom combination of abrupt onset, maximum intensity within seconds, and progressive encephalopathy raised the immediate concern of an aneurysmal subarachnoid hemorrhage evolving into intracranial hypertension. The clinical situation demanded immediate “neurological rescue” to document the time-sensitive evolution of secondary injury mechanisms and maximize the potential period for definitive intervention.

At the initial assessment, the patient was adequately drowsy but arousable to verbal stimulation. His Glasgow Coma Scale (GCS) score wandered between 12 and 13, primarily due to verbal output slowing, delayed response to stimulation, and poorer attentional capacity. His speech was intermittently incoherent, accompanied by initiation hesitation, and examples of thinking without a coherent context, all of which were attributable to early cortical disruption, meningeal irritation, and impaired cerebral autoregulation for cerebral blood flow. As the evaluation continued, there was also a noted dramatic reduction in his ability to sustain cognitive engagement—represented by incomplete sentences, arrest of thoughts in the middle of a thought, and gross behavioral disengagement—all suggesting further worsening of brain function.

The patient had considerable nuchal rigidity that was observed, and even a minor passive flexion motion of his neck also initiated an unprovoked withdrawal response, which suggested diffuse subarachnoid hemorrhage in the basal cisterns and posterior fossa regions. Kernig’s and Brudzinski’s signs were not formally elicited, as the patient was clearly uncomfortable; combined with the degree of rigidity, the findings were apparent enough to presume meningeal irritation and unstable/intracranial dynamic pressures. The preliminary examination of cranial nerves was overall intact; however, further analysis of the findings revealed subtle yet significant deficits present such as grossly symmetrical pupils, only sluggishly reactive but still reactive, indicating early diencephalic compromise; mildly impaired vertical gaze which indicated some mild dorsal midbrain involvement from either aqueductal obstruction or pressure from the posterior fossa; and delayed vestibulo-ocular responses, indicating compensated performance based on deficits in brainstem integrative circuits with a possible condition affecting performance such as ischemia in advanced deep perforator territories or some early obstructive hydrocephalus. Motor evaluation demonstrated preserved bulk and tone, preserved strength, impaired execution (i.e., slow initiation), and intermittent lapses in volitional motor control. Although there was no evidence of hemiparesis or focal weakness, subtle examples of dysdiadochokinesia and fatigable grip strength indicated possible evolving alterations to the subcortical motor circuits or cerebellar–brainstem interaction or conditions such as a nonserous hemorrhagic compression at the level of the cerebellopontine angle or early ischemia to the perforated vertebrobasilar branches. Within the context of encephalopathy, what was later noted in the context of cortical–subcortical deficit of function would render even the possibility of isolated tract pathology secondary.

The patient was clinically declining even while being observed and in real-time. Cognitive outputs were changing from delayed yet coherent, to monosyllabic outputs, to intermittent non-responsiveness requiring increasing amounts of external stimulation to engage him. What was increasingly noted in terms of clinical findings reflected increasing intracranial pressure (ICP), decreasing cerebral perfusion, and failure of autoregulatory reserve. However, one transitioned from a state of higher order dysfunction, through some transition period, to evident compromise to the function of the lower brainstem, urgently requiring a rescue level of care. The patient required emergent transfer to the neurosurgical intensive care unit for hemodynamic stabilization, continuous neurological status monitoring, and high-fidelity cerebral vascular imaging. Given the rapid speed of neurological decline, combined with a possible posterior circulation source, the threshold of intervention was low. Either way, we relied on a supreme-saturated, juxtaposed, and layered approach to mitigate secondary injury and preserve the surgical optionality.

In an effort to develop a management strategy on the basis of potential subsequent intervention, urgent digital subtraction angiography (DSA) imaging was performed within a matter of hours to locate the source of any hemorrhage as well as describe the imaging findings in the context of the vascular architecture and the ongoing risk the patient was under. Given the suspicion of the posterior fossa and the potential for increases in clinical decline, both traditional and high-resolution rotational axials of the vertebrobasilar axis were submitted to aid in the development of a management strategy.

The DSA findings (Figure 1) in the posterior circulation confirmed there was presence of a saccular aneurysm at the origin of the left PICA, at the vertebral artery take-off, measuring approximately 3 mm in width and size, with an irregular contour/character and focal pooling of contrast, which are key radiographic findings in suggesting recent rupture and wall instability. With regards to the vertebral artery, it had preserved flow characteristics and perfusion pressure; however, it was noted that the PICA followed an inferior-medial trajectory and a sharply acute angulation towards the lower brainstem, which, on that trajectory, affected the hemodynamics of both the injury and perfusion of surrounding structures, making it technically impossible to gain endovascular access and hemodynamically risky to adjacent perforators.

To further identify the aneurysm’s exact anatomy, orientation in three-dimensional space, and relation to important perforating branches, a three-dimensional rotational angiography (3D-DSA) was undertaken so that we could assess the aneurysm neck better and its integration with the PICA and vertebral artery. The reconstruction (Figure 2) verified that endovascular coiling was not a reasonable option due to the broad-necked configuration and acute vessel takeoff, which placed the PICA at greater risk of unintentional occlusion, which could lead to catastrophic brainstem and cerebellar infarction. There was also no sufficient landing zone for a stent (assisted technique) or flow diversion since any deployed device would lack adequate blood flow into the proximal portion of the PICA and present a significant ischemic risk.

Dynamic contrast angiography indicated abnormal brain perfusion patterns characterized by rapid opacification and delayed, poor outflow—consistent with intra-aneurysmal flow stasis and heightened risk of rupture. There was also poor collateralization to the PICA territory, which necessitated preserving antegrade perfusion. Sacrificing the parent vessel would have carried a very high risk of brainstem infarct, lower cranial nerve palsies, and posterior circulation ischemia.

Ultimately, it was decided to proceed with microsurgical clipping considering the patient’s clinical deterioration, the aneurysm morphology (making endovascular management difficult) and the intradural location of the diseased artery. We also weighed the option of doing a left, far-lateral craniotomy with partial C1 laminectomy, for direct visualization, proximal vascular control, and cerebellar retraction. This approach allowed a direct dissection and eventual placement of the clip to the aneurysm neck, while preserving continuous flow through the PICA.

The patient was placed on a park bench to maximize cerebellar relaxation and allow for any lateral trajectory. The head was secured in a three-pin Mayfield clamp, turned to the contralateral side, with slight flexion to expose the vertebral artery, with the origin of the PICA. A curvilinear incision was made posterior to the mastoid while extending inferiorally toward C1 for the removal of the suboccipital musculature for sufficient exposure. The suboccipital musculature was dissected in layers, and we took care not to injure the greater occipital nerve to limit post-operative neuralgia. The atlanto-occipital membrane was opened with a partial laminectomy at C1 with high-speed drilling under fluid irrigation employed to avoid injury to the venous plexus (other than a moving reference point) or destabilization of the C1-C2 complex. A far-lateral craniotomy was performed from the inferolateral occipital bone to the foramen magnum margin to create a focused but sufficiently large operable corridor.

The dura was opened, and controlled cerebrospinal fluid (CSF) was released from the cisterna magna to allow for cerebellar relaxation while we accessed microsurgically, without fixation and variable retraction. The dura was tacked laterally, and during sharp arachnoid dissection, the intradural vertebral artery was visualized and its bifurcation into the PICA and anterior spinal artery. We were able to identify a safe zone for proximal control that allowed us to temporarily occlude the vertebral artery if needed.

The aneurysm was very deep within the lateral medullary cistern, with arachnoid and brainstem adhesions, with deep perforators of posterior circulation anatomy surrounding the aneurysm neck. Dissection was performed within the suprahypoglossal triangle defined by the hypoglossal nerve and glossopharyngeal/vagus nerves (to allow for a slightly more controlled corridor to the aneurysm neck and to decrease traction on the cranial nerves). The vertebral artery was then mobilized away to further expand our working window. Sharp microsurgical dissection was performed cautiously to free the dome of the aneurysm without disturbing fragile perforators. The logic behind deploying a straight Yasargil titanium clip rather than a curved Yasargil titanium clip was purely based on the angulation of the aneurysm and the need to exclude it completely. At the beginning of the clipping, near complete obliteration was accomplished, but retrograde filling was observed on the lateral neck of the aneurysm. With causal instrumentation and counter-traction, we offered a smart re-adjustment and covered the residual flow utilizing a fenestrated clip; distal vessels were patent. On inspection, aneurysm obliteration was confirmed without any compromise of perforators and maximal hemostasis was reached. Dura was tightly closed with a suturable graft to provide an airtight closure to alleviate CSF leaks. Thereafter, the suboccipital musculature was re-approximated in layers. At closure, the field was dry with no residual aneurysm and no evidence of vasospasm was seen. Due to the complexity of dissection and decision-making that was warranted to optimize the position of the clip in the first instance, it was deemed prudent to closely monitor the patient post-operatively for any signs of perforator ischemia.

A number of intraoperative photographs were taken to depict the critical microsurgical procedural steps of dissection of an aneurysm and deployment of the clip. These photographs (Figure 3) show the anatomical exposure obtained via the lateral far approach, the careful engagement of the clip, and the final operative field. In addition, these figures show the process stepwise, from identifying the proximal vessel and safely dissecting around the critical neurovascular structures, to deploying the clip and confirming the exclusion of the aneurysm. The documentation reinforces the described microsurgical approach and reinforces the technical points to be mindful of to maintain PICA flow and arterial perforators to the brainstem when performing complex surgery on posterior circulation aneurysms.

The patient was then transferred to the neurosurgical ICU post-operatively. He was neurologically stable with intact brain stem reflexes, symmetric motor response, a GCS of 14, consistently and reliably following commands, and no new deficits; pupils were brisk and symmetric; extra-ocular movements were full; and facial function was maintained. There was mild brassy hoarseness without evidence of dysphagia or aspiration, which was deemed to be transient vagal neuropraxia.

Throughout the first emergent 24 h period post-operatively, the patient had an early non-contrast CT scan performed without concerns for hemorrhage, infarct or hydrocephalus (Figure 4). Using the axial and coronal views, we confirmed the appropriate placement of the clip as outlined and we could also confirm posterior fossa structures were preserved; at that point, ventricular size remained symmetrical, which implied stable CSF circulation without any adverse complications attributed to surgery.

The patient’s neurological status was stable over the first 24 h, and he showed progressive improvement in arousal, alertness, and attention. By post-operative day two, he was awake, oriented, communicative, remembered and understood, and was talking fluently. He had a normal motor exam and was symmetric in all extremities. He was able to sit without truncal instability, and early mobilization began. With assistance, he stood unassisted, with intact posture and proprioceptive control. He also performed cerebellar testing, which demonstrated no dysmetria, tremor, or gait ataxia, demonstrating that PICA provided cerebellar function was preserved.

Evaluation of swallowing revealed safe intake of oral secretions, and he began a phased re-introduction of enteral nutrition. The mild hoarseness noted after extubation decreased progressively, and had nearly resolved by post-operative day three, consistent with transient vagal neuropraxia likely caused by intraoperative traction rather than injury to the nerve itself. Lower cranial nerve function was intact, with verbalized praise for palatal elevation, midline protrusion of the tongue, and normal pharyngeal reflexes.

Because of the location and the independent PICA supplied perforators at the base of the aneurysms, neurological status was monitored closely for evidence of delayed ischemic changes. By post-operative day five, the patient was able to ambulate and walk with a steady gait, without evidence of disequilibrium or coordination issues. He tolerated a full diet, engaged with rehab staff, and did not appear fatigued nor exhibit cranial nerve dysfunction or hydrocephalus. He remained neurologically stable, so he was cleared to transfer from the intensive care unit to a neurosurgical step-down unit.

On post-operative day seven, a planned reassessment was performed, including repeat non-contrast cranial CT (Figure 5). The imaging demonstrated anatomical posterior fossa without mass effect or infraction, and no ventricular enlargement. The aneurysm clip had a stable position, with no indication of hemorrhage, edema, or disturbance in the flow or compression of the cerebrospinal fluid. The patient’s ventricle appeared symmetrical, and midline structures appeared intact. In summary, in the absence of delayed complications, the patient was discharged home in excellent neurological condition, with specific rehabilitation and follow-up planned.

At discharge, the patient had a final CT scan to confirm definitive stability at the operative site and to exclude any diseases that would have developed late and could have a negative impact on long-term outcome (see Figure 6). The axial view confirmed an anatomically preserved posterior fossa. There was no evidence of infarction, hemorrhage, or delayed hydrocephalus. The cerebellum appeared well-structured. There was no evidence of parenchymal atrophy/degeneration or non-structural distortion. In the coronal view, the aneurysm clip remained stable and the scans were otherwise negative for any secondary mass effect or delayed vasculospasm. The ventricular system was unchanged and there were no dilated or asymmetrical ventricles suggesting problems with CSF flow. Neurologically, the patient was fully independent, had no residual deficits and was functionally at baseline walking independently and without assistance, on a full diet, with intact speech and cognitive function and coordination. Based on his exceptional recovery and the findings of stability from imaging, he was discharged home with a follow up angiographic assessment scheduled for one month to ensure definitive aneurysm exclusion and to see if there was any vascular remodelling at the clip site. With a structured long-term care plan, stable neurological examination and an optimal functional outcome, his prognosis was excellent and he would likely make a full neurological recovery with no significant long-term consequence.

This case illustrated the methodical approach to managing a ruptured PICA aneurysm using a structured microsurgical approach. The patient ultimately had a successful outcome with no post-operative complications and was discharged having a stable neurologic exam. This case illustrates the benefit of a tailored surgical approach, and appropriate peri-operative management for these complex aneurysms in the posterior circulation.

## 3. Discussion

Managing posterior inferior cerebellar artery aneurysms remains a complex challenge of cerebrovascular surgery, which necessitates a careful compromise between durable exclusion of aneurysms and preservation of neuro-vascular structures. In terms of treatment, anterior circulation aneurysms possess a variety of clear treatment paradigms; however, PICA aneurysms are unique due to their technical and biological difficulties associated to their deep locations, morphologies that may vary between individuals and their proximity to critical brainstem perforators [18]. While endovascular techniques have changed the dynamics of treatment interventions of many intracranial aneurysms, PICA aneurysms are still among the most challenging and underutilized via an endovascular approach due to anatomic and hemodynamic challenges, with intervention by microsurgery often the preferred option [19]. We hope this case report adds to the growing body of literature that recognizes the importance of individualized, anatomy-driven surgical tactics to prompt optimal outcomes for these rare but potentially devastating lesions. This ongoing discussion promotes a normalizing of the paradigm of cutting-edge insight into the global epidemiology, hemodynamics, comparative-therapeutic strategies, genetics, and future technologies in order to refine practical understanding of PICA aneurysms and discover new and exciting pathways to inspire research and influence clinical practice.

The global distribution and treatment preferences of this aneurysm spotlights a tremendous regional variation and highlights the differences in both case volume, surgical capability, institutional practices, and accessibility to endovascular treatment options. The large aneurysm registries existing both in North America and Western Europe demonstrate a significant shrinkage in microsurgical treatment approaches, with more than 85% of posterior circulation aneurysms being treated endovascularly [20,21]. In contrast, high volume cerebrovascular centers in Asia are still emphasizing microsurgical clipping for complex aneurysm morphologies, especially those with wide necks, high angulation, or involving the parent vessel [22]. This difference is not only related to technological availability, but rather, longitudinal outcomes data suggest that surgical treatment is still superior for those with high recurrence risk. In addition, genetic and epidemiological studies are also suggesting that the patterns of vascular remodeling may differ by population, as East Asian cohorts have shown a larger prevalence of vertebrobasilar dolichoectasia that can contribute to increased hemodynamic stress at the vertebro–PICA junction [23]. These findings underscore the need for a personalized treatment paradigm that uses a why and how procedural platform but also factors in both patient specific anatomical factors and population based epidemiologic data [24,25].

Recent developments in computational fluid dynamics have shed important light on the unique hemodynamic forces influencing PICA aneurysm formation, growth, and rupture. In contrast to bifurcation aneurysms caused by higher velocity flow impingement, PICA aneurysms form in a location of low flow turbulence that leads to cumulative endothelial damage, progressive wall remodeling, and localized inflammation [26]. Furthermore, hemodynamic modeling has shown that the irregular distributions of wall shear stress on PICA aneurysms make them disproportionally rupture-prone at smaller sizes even compared to anterior circulation aneurysms [27]. Large population cohort studies have confirmed that small posterior circulation aneurysms have a significantly higher risk of rupture than similar sized anterior circulation aneurysms, which challenges existing size-based treatment thresholds. Histopathological studies of PICA aneurysm walls have shown a greater incidence of adventitial fibrosis, formation of mural thrombus, and vasa vasorum proliferation, indicating a different underlying molecular basis for their aggressive rupture characteristics [20]. These studies further justify the use of high-resolution imaging technologies such as vessel wall MRI and four-dimensional flow MRI that could provide further differentiation in the rupture-risk stratification process and further help guide surgical treatment options for unruptured PICA aneurysms [20,28].

Histopathological studies of PICA aneurysm walls have shown a greater incidence of adventitial fibrosis, formation of mural thrombus, and vasa vasorum proliferation, indicating a different underlying molecular basis for their aggressive rupture characteristics [29]. These studies further justify the use of high-resolution imaging technologies such as vessel wall MRI and four-dimensional flow MRI that could provide further differentiation in the rupture-risk stratification process and further help guide surgical treatment options for unruptured PICA aneurysms [30]. Large retrospective studies have demonstrated a recurrence rate of up to 40% for endovascular coiling of PICA aneurysms, with nearly 25% undergoing retreatment within five years due to coil compaction or incomplete occlusion [31]. Similarly, while flow-diverting stents can be successful in large anterior circulation aneurysms, they have not been widely used in PICA aneurysms due to their unpredictable hemodynamic effects and high rates of vessel thrombosis [31]. These limitations support the continuing role of microsurgical treatment in cases where durable aneurysm exclusion and PICA flow preservation are crucial for long-term neurological functional status. The following table (Table 1) aims to provide a summary comparison of the most frequently cited data on PICA aneurysm treatment, in order to address population and epidemiological trends, biomechanical neurosurgical principles, and eventual future treatments.

Microsurgical clipping remains the standard of care for complex PICA aneurysms because it provides immediate, complete, secured aneurysm exclusion with a long-term patency of the involved vessels [36]. As described above, the far-lateral approach remains the gold standard for vertebro–PICA aneurysms because it directly exposes the aneurysm, provides the ability to control the proximal vertebral artery, and also reduces cerebellar retraction [14]. However, with the refinement of the microsurgical approach, there is considerable opportunity to improve the safety and efficiency of surgery. Techniques such as utilizing partial condylar resections, carefully strategizing C1 laminectomy, optimization of dural openings, and better exposure ultimately allowed us to expand our microsurgical toolkit while decreasing cranial nerve injury [47]. This case demonstrates the benefits of a modified far-lateral approach based on the unique anatomy of the aneurysm, utilizing specific microdissection techniques to facilitate clip placement and maintain flow into the distal PICA. When compared to previously reported cases, this aneurysm had an unusually high angulation and was located deep in the lateral medullary cistern, requiring a more advanced dissection plan to achieve complete occlusion while preserving perforators.

Genetic components to the development of intracranial aneurysms comprise an expanding research area, providing additional understanding surrounding the increased risk profile of specific populations that have higher prevalence of posterior circulation aneurysms, as well as why specific sites of aneurysm, including the PICA, are preferentially ruptured. Large-scale genome-wide association studies (GWAS) have led to the identification of several genetic loci related to aneurysm susceptibility, including variants in genes encoding elastin (ELN), collagen type I alpha-2 (COL1A2), lysyl oxidase (LOX), and matrix metalloproteinases (MMPs) [48]. The identification of mutations in SOX17, a transcription factor necessary for endothelial stability and vascular integrity that was strongly associated with posterior circulation aneurysms, was one of the most significant discoveries. Additional over-expression of MMP-9 and IL-6 seen in aneurysm wall tissue was also correlated with a higher rupture risk, confirming the role that chronic inflammation also had in the development of aneurysms [49,50]. Perhaps the most exciting aspect of the influence of genetic loci and polymorphisms on the development of cerebral aneurysms was the evidence of epigenetic changes, including but not limited to, DNA methylation and histone acetylation changes in extracellular matrix remodeling genes pointed toward the reality that it is not just genomic susceptibility that predisposes individuals to the development of cerebral aneurysms but environmental and epigenetic changes that further implicate the risk [51]. Overall, the identification of simple genetic molecular mechanisms combined with complex environmentally mediated changes of chronic inflammation and epigenetics has significant implications on future aneurysm risk prediction and treatment, individualized to patients’ aneurysms using negative-modulating pharmacy-guided invasive treatment or an anti-inflammatory treatment being able to stabilize preoperative aneurysm walls to limit their risk of rupture [52].

The management of PICA aneurysms will most likely be defined by hybridized microsurgical-endovascular approaches combined with available advanced intraoperative imaging and artificial intelligence (AI) to assist in risk prediction models. Current works introducing machine-learning processes that have been trained on large aneurysm databases are demonstrating great accuracy at predicting risk of rupture, risk of growth of the aneurysm, and likelihood of recurrence after treatment [53]. AI-driven intraoperative surgical systems are being developed to utilize intraoperative imaging to maximize the clip sittings for secure aneurysm exclusion, as well as to identify and “mark” high-risk perforators during surgery to avoid catastrophic consequences. Development of miniaturized flow-diverting stents to treat small-caliber arteries (including the PICA) is also in early-stage trial research and will be earmarked as a specific treatment for a selected few PICA aneurysms [54].

AI-assisted image segmentation and rupture-risk modeling are now able to identify small but consequential morphological and hemodynamic features (e.g., dome irregularity, inflow jet location, angulation, and flow stasis) that indicate structural instability in anatomically demanding posterior circulation aneurysms like vertebro–PICA lesions. In this patient, the deep location, torrential vertebral origin, and delayed intra-aneurysmal outflow represented a high-risk combination, which would benefit from predictive models. When coupled with high-resolution angiography and microsurgical experience, a successful AI-assisted preoperative analysis has the potential to strengthen surgical planning, inform the clip trajectory, and maximally protect perforators in a limited operative field.

Surgical treatment of PICA aneurysms is a difficult endeavor requiring great anatomical understanding and flexibility because of the anatomy’s complicated five-segment structure, consisting of an anterior medullary, lateral medullary, tonsillomedullary, telovelotonsillar, and cortical section. Each segment presents its challenges. Aneurysms at the VA–PICA junction (e.g., our case), which typically nestle inside the lateral medullary cistern, present a unique navigation challenge, as this region has a lot of eloquent brainstem perforators and cranial nerves IX–XII to consider [14]. In addition, surgical exposure and proper clamp placement likely need an exquisitely delicate approach to prevent ischemic infraction and permanent injury to neighboring nerves. The tight spaces of the cistern and the low ventromedial angles complicate maneuvers and often require skull base adjustments to safely navigate the surgical site. Alternatively, aneurysms in the telovelotonsillar segment include a pattern of distorted anatomy because of previous hemorrhage, which can hinder access and increase the risk of venous infarction when retracting the tissues [55].

Not all PICA aneurysms present in a saccular format. Because of their distal segments, fusiform, dissecting, or dolichoectatic aneurysms more often than not result in a dilemma of surgical treatment, as clipping and coiling will not provide sustained exclusion of blood flow without overwhelming risk of disruption [56]. There are also mechanical limitations with flow diversion methods that rely on the endovascular techniques of a parent artery. Acute angulation of the PICA origin, the diminutive outer dimensions of the artery, and the preponderance of perforators are often inflexible enough to prevent safe catheterization or retention of coil in many cases. Recently, there has been a limited number of low-profile intraluminal flow-dividers entering the clinical trial process for selectively designed small-vessel devices; however, they are not a substitute for use in most proximal PICA due to the risk of acquiring a medullary infarct [57].

In these surgically hostile environments, revascularization development is often essential in pursuing functional preservation. One of the most elegant options available is the side-to-side PICA–PICA bypass, which does not require graft harvesting and takes advantage of the symmetrical perfusion of vascular territory. When the contralateral anatomy is unfavorable, OA–PICA bypass with short-segment interposition grafts (i.e., radial artery and saphenous vein) can recreate flow while facilitating sacrifice of the parent artery [58]. Additionally, flow-enhancing bypasses that aim to supplement perfusion but do not fully replace flow but can offer a hybrid solution in some patients with incomplete collaterals. So that it is clear, it is important for us to acknowledge that intraoperatively the use of dual-fluorescence indocyanine green videoangiography, micro-Doppler velocimetry, and real-time quantitative flow analysis (Q-Flow) has made objective evidence of perfusion at various time points before, during and after occlusion of the aneurysm a reality and has diffused the guesswork of microsurgery decision making, particularly in high stakes situations. Future perspectives will combine computational hemodynamic modeling, intraoperative perfusion mapping, and AI image registration to simulate pre-surgical flow-diversion or bypass modes and allow for tailored strategic development of the aneurysm phenotype [59,60]. As the technology matures and is viewed along with advancing microsurgical expertise, we are likely witnessing the next evolution of posterior fossa aneurysm management, especially in high-risk areas like PICA [61].

This case reiterates that microsurgical clipping is indeed a safe and durable treatment for complex PICA aneurysms, specifically where endovascular options are limited, due to anatomical obstacles. The successful exclusion of an aneurysm, with no post-operative complications, and full neurological recovery suggests that precise microsurgical planning, refined intraoperative microsurgical techniques, and structured post-operative management allowed this case to be successful amidst all the unprecedented challenges. The field of cerebrovascular treatment continues to evolve, but integrating advanced technological innovation with surgical expertise should serve to enhance the precision, safety, and efficacy of PICA management so that further patients may receive the most sophisticated, personalized, and individualized care possible.

## 4. Conclusions

Details surrounding the management of posterior inferior cerebellar artery aneurysms will continue to evolve in the context of surgical complexity, endovascular advancements, hemodynamics, and evolving molecular concepts. Using a pretty broad definition, although there has been an explosion of minimally invasive treatment options for many intracranial aneurysms, PICA aneurysms are among the most technically challenging to treat, requiring a patient-specific approach that considers the inherent anatomy, durability of treatment, and functional preservation. This case supports the notion that microsurgical clipping remains a viable method of definitive treatment, especially in the instance of PICA aneurysms, where there may be endovascular limitations such as recurrence, incomplete occlusion, and compromised blood flow to the parent artery.

As cerebrovascular surgical techniques continue to advance, the role of long-term monitoring should be optimized to improve outcomes for PICA aneurysm patients. As discussed above, the unique profile of recurrence rates for posterior circulation aneurysms will require a formalized process for follow-up. DSA should remain the gold standard for immediate follow-up, but magnetic resonance angiography (MRA) or computed tomography angiography (CTA) may be more appropriate for long-term follow-up of these angiographic anomalies. Although there are risks of delayed ischemic complications, the risks associated with delayed ischemic complications from perforators can be theoretically minimized with meticulous microsurgical technique, which highlights the importance of post-operative perfusion imaging to understand the dynamics of blood flow for the brainstem and cerebellum over time. In addition to radiographic follow-up, the role of neurocognitive and functional recovery of neurological status during the follow-up time should be considered in the post-operative evaluation, since posterior circulation aneurysms, especially those involving the PICA, may lead to vestibular dysfunction, subtle deficits in motor coordination, and impacts on quality of life that go beyond mere exclusion of the aneurysm. In addition to technology, we are seeing genetic and molecular studies transform our understanding of aneurysm pathophysiology and create more individualized treatment paradigms. Recently, genome-wide association studies have identified mutations in SOX17, ELN and COL1A2 associated with aneurysms. These mutations are more common in posterior circulation aneurysms, where the arterial remodeling of the wall and endothelial dysfunction are more salient features. Furthermore, it has been documented that, overwhelmingly, inflammatory markers like MMP-9 and IL-6 were devoid of precursor development in aneurysm wall tissue; this also suggests a more prominent role of chronic inflammation in aneurysm development and the potential for rupture risk. This means that aneurysm treatment may not only be surgical or endovascular but also could consider targeted molecular therapies to stabilize the aneurysm walls with pharmacological inhibition of inflammatory cascades or vascular extracellular matrix degradation. While these possible therapies are in early investigative studies, the future of aneurysm treatment could differ from surgical or endovascular treatment to treat aneurysm behavior before they rupture.

With the speed of innovation increasing, AI, robotics and augmented reality will drive the transformation of decision-making and procedural execution in cerebrovascular surgery. AI-based rupture prediction models on large aneurysm datasets reliably distinguish patient-based risk factors, incorporate hemodynamic patterns of flow, vessel wall integrity and genetic markers to provide optimal treatment selection. Augmented reality-assisted microsurgery is coalescing into a unique tool for improving intraoperative visualization and allowing surgeons to augment the surgical field to achieve real-time imaging that can improve the placement of clips and preservation of perforators. Robotic-assisted microsurgical platforms, built specifically to decrease tremor and provide submillimeter accuracy, may build on this even more in these difficult cases of deep-seated aneurysms such as those seen in the PICA territory. The current advancement of miniaturized flow-diverting stents to treat small-caliber vessels remains an area of excitement, although the risk of thrombosis as well as unpredictable patterns of flow redistribution remain considerable barriers to their use in PICA aneurysms.

While technology continues to push the field forward, ethical and practical challenges exist, demonstrating equitable access to cutting-edge interventions. First, the development of AI-enhanced rupture prediction models and robotic microsurgery comes with its own challenges within healthcare access, as these technologies will likely be delivered by high-resource centers and not blur so many of the gaps within neurovascular care. Second, as we diverge from our previous treatment paradigms, there is a discussion about whether to justify a shift in aneurysm care toward a model more akin to center-based specialization. Doing so might direct complex cases such as broad-necked PICA aneurysms to centers experienced in both microsurgery and advanced endovascular techniques, but it would at least in part function as a way to help address the volume issues that we will return to in treatment considerations. The fundamental balancing act of the job will always remain a tension between access to care regionally and the need to optimize outcomes ultimately for patients. Another pressing issue in cerebrovascular research is how best to deal with small, unruptured PICA aneurysms from here on out. The historical emphasis on the size of the aneurysm as a risk factor for rupture is increasingly being challenged by hemodynamic and genetic research telling us that posterior circulation aneurysms, especially the ones stuck in low-flow, high-shear stress areas like PICA, rupture at disproportionately greater rates than anterior circulation aneurysms of the same size. As computational hemodynamic models continue to advance, we may see a shift in paradigms toward earlier and more aggressive management of unruptured aneurysms based on the following: high-risk unruptured PICA aneurysms with abnormal flow dynamics, or aneurysms with wall enhancement on vessel wall MRI, or genetic variants of the patient relating to instability. This shift would represent a refinement of the treatment thresholds, which includes the incorporation of multi-factorial models of risk stratification that extend beyond size alone.

As treatment paradigms evolve, this case shows the important balance between an anatomy-driven, patient-specific approach and measured decision-making in the management of posterior circulation aneurysms. Technology is certainly ever evolving and will be a shining influence on the future of neurovascular surgery; however, robust decision making, a precise technique, with a willingness and respect for the innate complexity of cerebrovascular anatomy, will never lose relevancy. Ongoing refinement in the cerebral space of surgical and endovascular techniques and also advances in the biology of aneurysms will ensure the field continues to advance so we can deliver treatment that is safer, more efficacious and more individualized to the biological attributes of each patient and their disease.

The future of management of PICA aneurysms will not fall victim to a single approach but instead may build on an interdisciplinary alliance of microsurgical skills, endovascular modification and computational product. With advancing AI improving risk analyses and surgical techniques, robotics delivering more precise surgical techniques and novel molecular therapies demonstrating critical aneurysm-stabilizing properties, the future, with a flourish of new technologies, will usher in a new era in cerebrovascular interventions. While this case report is coming to a conclusion, the vision for improving aneurysm care will continue, a progressive exploration where every advancement of a technology, every step toward a new surgical technique, and every scientific innovation will contribute to improved individualized and safe treatment for our future patients.

## Figures and Tables

**Figure 1 jcm-14-05374-f001:**
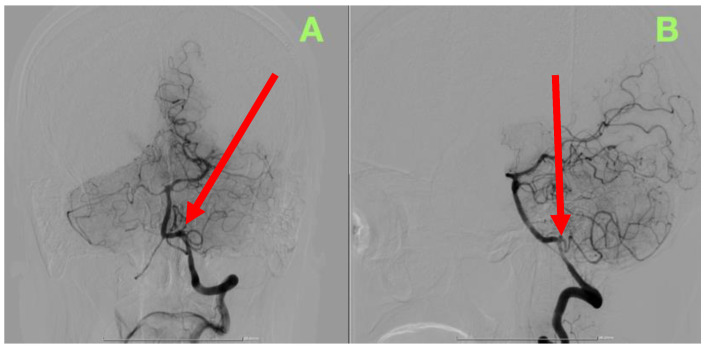
Preoperative DSA. (**A**) Anteroposterior projection showing the left vertebral artery and posterior circulation with a saccular aneurysm at the PICA origin (red arrow). (**B**) Lateral projection demonstrating the aneurysm’s broad-necked morphology and its close relationship to the brainstem.

**Figure 2 jcm-14-05374-f002:**
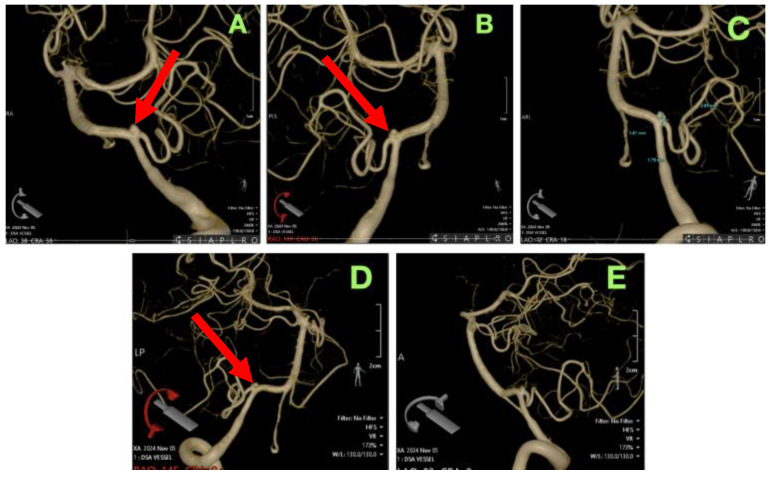
The 3D-DSA of the left vertebral artery providing an enhanced spatial representation of the aneurysm and its critical vascular relationships. (**A**,**B**): Oblique and lateral projections revealing the aneurysm’s fusiform configuration and its sharp angulation at the origin of the PICA (red arrow), emphasizing the anatomical complexity of the parent vessel’s course. (**C**): Detailed morphometric analysis displaying precise measurements of the aneurysm’s neck width (1.79 mm), dome diameter (2.61 mm), and proximal parent artery caliber, critical for determining microsurgical clip selection and placement strategy. (**D**,**E**): Additional rotated perspectives illustrating the aneurysm’s (red arrow) three-dimensional orientation within the vertebrobasilar system, demonstrating its deep-seated position in the lateral medullary cistern, its adjacency to brainstem perforators, and its limited access corridor, which played a pivotal role in selecting the far-lateral approach for definitive intervention.

**Figure 3 jcm-14-05374-f003:**
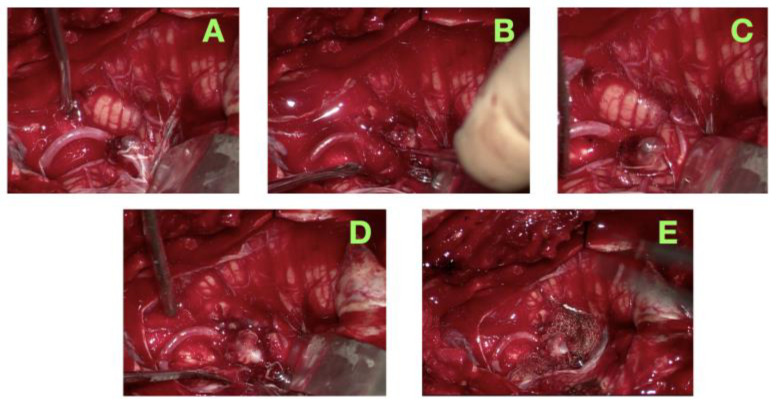
Intraoperative microsurgical exposure and clipping of the ruptured PICA aneurysm. (**A**): Exposure of the intradural vertebral artery and identification of the PICA origin and saccular aneurysm at the VA–PICA junction. (**B**): Close-up view during precise microdissection around the aneurysm dome with preservation of adjacent perforators. (**C**): Trial clip application to evaluate neck angulation and maintain parent vessel patency. (**D**): Final clip positioning achieving complete aneurysm exclusion with preserved PICA flow. (**E**): Final operative field demonstrating hemostasis, intact regional anatomy, and absence of perforator compromise.

**Figure 4 jcm-14-05374-f004:**
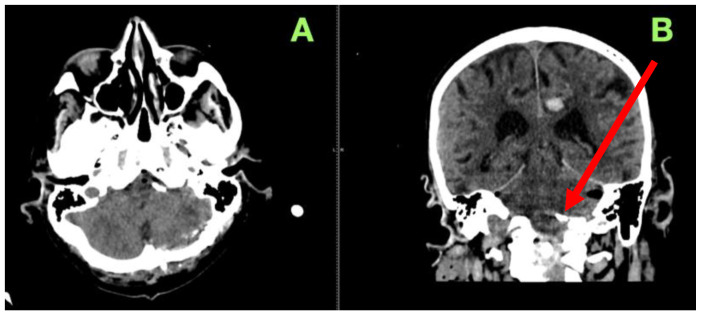
Immediate post-operative non-contrast computed tomography scan. (**A**): Axial view confirming no post-operative hemorrhage, ischemia, or hydrocephalus, with well-preserved posterior fossa structures and no mass effect on the brainstem. (**B**): Coronal view demonstrating adequate aneurysm clip positioning (red arrow), no vascular compression, and stable ventricular configuration, ruling out CSF circulation impairment. A small parafalcine hyperdensity is visible on the left frontal convexity, likely a clinically silent contrecoup hemorrhage unrelated to the surgical site.

**Figure 5 jcm-14-05374-f005:**
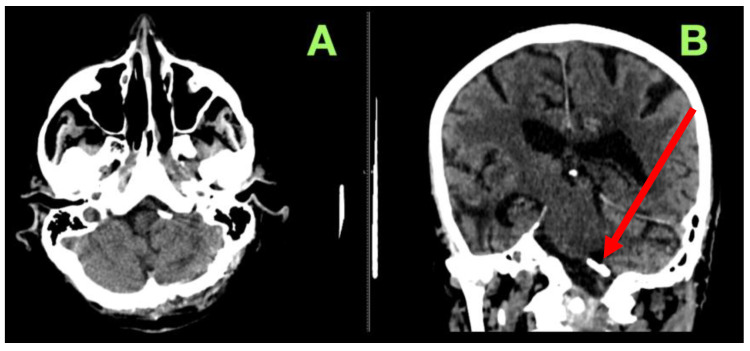
Non-contrast computed tomography scan at post-operative day seven. (**A**): Axial view demonstrating a well-preserved posterior fossa without mass effect, infarction, or hydrocephalus, confirming optimal surgical outcomes. (**B**): Coronal view showing stable ventricular morphology, no evidence of delayed ischemic injury, and complete exclusion of the clipped aneurysm without secondary complications (red arrow).

**Figure 6 jcm-14-05374-f006:**
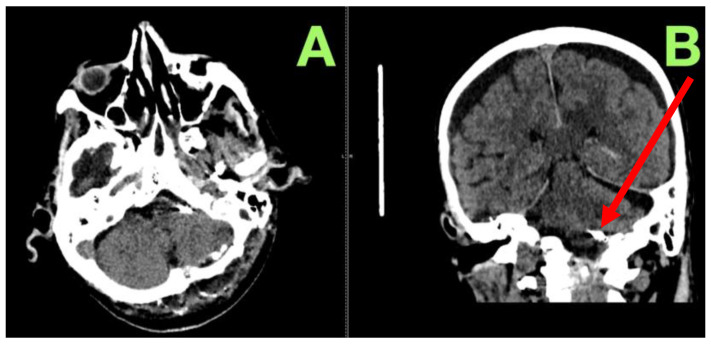
Non-contrast computed tomography scan at hospital discharge. (**A**): Axial view showing a structurally intact posterior fossa with no evidence of infarction, hemorrhage, or hydrocephalus. (**B**): Coronal view confirming stable aneurysm clip positioning (red arrow), preserved CSF flow, and no late-stage post-operative complications. Mild bilateral extra-axial fluid-density collections are present, interpreted as benign post-operative changes related to CSF redistribution following cisternal opening and brain relaxation. These findings remained clinically silent and resolved spontaneously during follow-up, with no impact on the patient’s recovery.

**Table 1 jcm-14-05374-t001:** Presents a highly detailed, structured synthesis of the most significant advancements in PICA aneurysm research and management, summarizing key findings from epidemiology, biomechanics, treatment strategies, genetics, post-operative outcomes, and future innovations. By integrating data from clinical trials, hemodynamic modeling studies, and genetic research, it provides a comparative framework for evaluating the advantages and limitations of both microsurgical and endovascular approaches, while identifying gaps in current treatment paradigms.

Category	Key Findings	Scientific Implications	Advantages	**Limitations and Risks**	**References**
Global Epidemiology and Regional Treatment Trends	PICA aneurysms account for 0.5–3% of all intracranial aneurysms. Higher prevalence in Asian populations, possibly due to genetic predisposition and increased screening. Microsurgical clipping remains more common in Asia, while endovascular coiling is preferred in North America and Europe.	Regional treatment disparities highlight the need for individualized approaches rather than a universal treatment paradigm. Genetic predispositions may play a role in regional aneurysm formation patterns.	Clipping provides definitive occlusion with lower recurrence rates. Coiling is minimally invasive with faster recovery.	High recurrence rates for coiling (30–40%), need for long-term follow-up, and risk of cranial nerve deficits post-clipping.	[32,33]
Hemodynamic and Biomechanical Risk Factors	PICA aneurysms develop in low-flow, high-shear stress regions, making them prone to rupture at smaller sizes. Flow stagnation, turbulent hemodynamics, and irregular wall shear stress increase rupture risk. Vessel wall MRI studies confirm higher prevalence of intra-aneurysmal thrombus and endothelial dysfunction in PICA aneurysms.	Small PICA aneurysms should not be observed conservatively based on size alone. Flow analysis using 4D-flow MRI and CFD modeling is increasingly used for risk assessment.	Advanced imaging (CFD, 4D-flow MRI) allows early rupture prediction.	High variability in patient-specific hemodynamics, difficult to establish universal treatment thresholds.	[34,35]
Microsurgical Clipping vs. Endovascular Coiling	Clipping achieves complete occlusion in 90–95% of cases with low recurrence. Coiling has a 30–40% recurrence rate, with 20–25% requiring retreatment. Stent-assisted coiling remains challenging due to small PICA caliber and high perforator risk. Flow diversion remains experimental for PICA aneurysms.	Microsurgical clipping remains the gold standard for broad-necked and deep-seated PICA aneurysms. Endovascular therapy is preferred for morphologically favorable aneurysms. Hybrid approaches (coiling before clipping) are being investigated.	Clipping is definitive with low recurrence. Coiling is less invasive, shorter recovery.	Clipping requires skilled microsurgical expertise, risks cranial nerve injury. Coiling has higher recurrence rates.	[36,37]
Genetic and Molecular Pathophysiology	SOX17, ELN, COL1A2, LOX mutations linked to posterior circulation aneurysms. Increased expression of MMP-9 and IL-6 correlates with aneurysm instability. Epigenetic modifications, such as TIMP3 hypermethylation, reduce vessel wall integrity.	Genetic screening may allow for early identification of high-risk individuals. Molecular-targeted therapies (MMP inhibitors, endothelial stabilizers) are in development.	Could enable personalized treatment strategies and early intervention.	Gene–environment interactions remain poorly understood. Targeted pharmacotherapy is not yet validated.	[38,39]
Post-operative and Long-Term Outcomes	Hydrocephalus occurs in 10–15% of ruptured PICA aneurysms, often requiring CSF diversion. Cranial nerve dysfunction (CN IX, X) occurs in 20–30% of cases post-clipping, but most resolve with rehabilitation. Microsurgical clipping offers better long-term occlusion than coiling.	Structured post-operative monitoring and early hydrocephalus detection are critical. Functional recovery is improved with early speech and balance rehabilitation.	Clipping provides lower recurrence, higher durability.	Post-operative cranial nerve deficits, risk of dysphagia and aspiration.	[40,41]
Hemodynamic Evolution and Flow Dynamics Post-Treatment	Coiling achieves immediate flow disruption but incomplete packing can lead to delayed aneurysm recanalization. Microsurgical clipping results in immediate, complete exclusion, eliminating flow-related aneurysm growth. Post-treatment flow alterations can induce unpredictable cerebrovascular resistance changes, requiring long-term follow-up.	Flow quantification using 4D-phase contrast MRI may help predict long-term hemodynamic changes post-treatment.	Allows real-time assessment of aneurysm recurrence.	High cost of imaging, unclear threshold for re-intervention.	[42,43]
Future Directions and Technological Innovations	AI-based aneurysm rupture prediction models integrating aneurysm morphology, wall shear stress, and genetic markers are under development. Augmented reality-assisted microsurgical navigation is being explored for improved clip positioning and perforator preservation. Miniaturized flow-diverting stents for small-caliber arteries remain experimental due to high thrombosis risk.	The future of PICA aneurysm management will involve hybrid surgical–endovascular strategies, real-time intraoperative imaging, and AI-driven risk stratification.	AI-driven analytics could allow for personalized treatment based on risk factors.	High costs, need for multi-left validation before widespread adoption.	[44,45,46]

## Data Availability

The data presented in this study are available on request from the corresponding author.

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
