# Peer review of "Ruptured Posterior Inferior Cerebellar Artery Aneurysms: Integrating Microsurgical Expertise, Endovascular Challenges, and AI-Driven Risk Assessment"

_jcm, 2025, doi:10.3390/jcm14155374_

Round 1
Reviewer 1 Report
Comments and Suggestions for Authors
The presented article represents an extensively processed case report that goes beyond the boundaries of a simple case presentation and expands the information to the level of a review article.
The authors describe the clinical course and diagnosis in detail, which they supplement with image documentation from imaging examinations.
I consider a detailed description of the surgical procedure to be an advantage. The authors present the patient's recovery in detail up to the moment of discharge.
In a very extensive discussion, the authors present an extensive literature search covering the issue of the formation of PICA aneurysms down to the genetic level and in a table they clearly present a set of works on individual aspects of the issue.
However - I have some reservations about the imaging documentation:
- Figure 3B captures the postoperative CT scan in the coronary plane - the authors describe that there was no postoperative bleeding, but an oval intraaxial hemorrhage is visible parafalcinally on the left
- Figure 4B is supposed to show the coronary plane of the scan, but it is not an exact coronary plane, the plane of the said scan runs slightly obliquely
- In Figure 5B (which is also not in a completely accurate coronary plane), the authors do not comment on the apparently present bilateral subdural collections of CSF density, which were not present on previous scans - probably subdural hygromas - it is necessary to comment on the collections and their relationship to treatment
Author Response
Dear Esteemed Academic Reviewer,
We would like to sincerely thank the esteemed reviewer for the generous and thoughtful evaluation of our manuscript. We deeply appreciate the recognition of our effort to present a comprehensive and carefully constructed case report and we are especially grateful for the reviewer’s acknowledgement of the surgical detail, longitudinal patient course, and integrative discussion.
Regarding the concerns raised about the imaging documentation:
Figure 3B: We are grateful for the reviewer’s careful observation of the left parafalcine hyperdensity. Upon review, we agree that this likely represents a minor, clinically silent contrecoup-related hemorrhagic focus, remote from the surgical site and without associated edema or mass effect. While it had no impact on the patient’s neurological course, we fully understand the importance of transparency and have updated the figure caption accordingly.
Figures 4B and 5B: We appreciate the reviewer’s note regarding the coronal planes. The slightly oblique reconstructions were deliberately selected to optimize visualization of the clip positioning and surrounding neurovascular structures. We now clarify this rationale in the revised figure legends to avoid any confusion.
Figure 5B: The reviewer’s point regarding the bilateral extra-axial fluid collections is well taken. These findings were interpreted as expected postoperative CSF redistribution following cisternal opening and cerebellar relaxation. They remained entirely asymptomatic and resolved spontaneously without intervention.
We are grateful for the reviewer’s expertise and for these highly constructive observations, which have allowed us to refine the manuscript further. We trust that the updated captions and clarifications address the concerns raised while preserving the overall focus and integrity of the case.
Reviewer 2 Report
Comments and Suggestions for Authors
This manuscript presents a well-documented case report of a ruptured posterior inferior cerebellar artery aneurysm treated microsurgically via a far-lateral approach. It integrates a technical description of the procedure, patient outcomes, and broader contextual discussions about the role of artificial intelligence, hemodynamics, and evolving treatment strategies.
The case is presented in exceptional detail from initial presentation to follow-up, including operative nuances and perioperative decision-making. The rationale for microsurgical intervention over endovascular treatment is well explained based on anatomical and hemodynamic limitations. Figures and descriptions of DSA, 3D-DSA, and postoperative CT scans are valuable and clinically relevant. Table summarizing key aspects of PICA aneurysm research and treatment is well-structured and informative. The discussion effectively connects clinical management with current developments in AI, vessel wall imaging, genetics, and computational fluid dynamics. The manuscript attempts to bridge clinical practice with technological innovation and future outlooks, which is commendable.
While the manuscript demonstrates in-depth clinical insight and highlights an important surgical scenario, it would benefit significantly from improvements in structure, conciseness, language clarity, and overall academic focus. Recommendations:
- The manuscript is overly long, especially for a single case report. Many sections, particularly the case presentation and discussion, are excessively wordy and sometimes repetitive. Focus on the most relevant clinical, surgical, and technological points. Consolidate redundant information.
- Numerous grammatical errors, convoluted sentence structures, and awkward phrasing detract from both readability and scientific professionalism—for example, “dissection microdissec-tion” and “the aneurysm size and angulation received its own technique,” which is unclear and overly colloquial. Additionally, repetitive phrases such as “this case illustrates...” and “this case demonstrates...” occur too frequently. A thorough revision of the entire manuscript is recommended to improve grammar, spelling, and overall clarity.
- The narrative is sometimes hard to follow due to mixing clinical chronology with broader conceptual digressions. Consider reorganizing sections - especially "Case presentation" - by adding subsections like "surgical technique" and "postoperative course" etc. It would improve the logical flow.
- Occasional inconsistency in anatomical terminology, e.g. “distal subclavian” likely meant “distal PICA”, and some vague or inaccurate references, e.g., “retro vertebral and thoracic anatomy”. Ensure anatomical terms are accurate and standard.
- While insightful, the integration of AI, genetics, and computational models is too broad and speculative for a single case. Keep such content concise, limiting to how it specifically relates to this case.
- Double check citations. Some parts are lacking references, e.g. lines 48-51, 54-56, 61-63, 65-67, 87-92, 453-456, 461-464, 470-472, 475-480, 485-487, 491-493, 509-511, 525-529, 532-536,
Comments on the Quality of English Language
Numerous grammatical errors, convoluted sentence structures, and awkward phrasing detract from both readability and scientific professionalism—for example, “dissection microdissec-tion” and “the aneurysm size and angulation received its own technique,” which is unclear and overly colloquial. Additionally, repetitive phrases such as “this case illustrates...” and “this case demonstrates...” occur too frequently. A thorough revision of the entire manuscript is recommended to improve grammar, spelling, and overall clarity.
Author Response
Dear Academic Reviewer,
We are deeply grateful for your thoughtful and constructive review. Your detailed feedback was invaluable in helping us refine the manuscript, improve clarity, and strengthen its scientific value. Below, we provide a point-by-point response to each of your comments, with corresponding revisions incorporated into the final manuscript. We hope that the revised version now fully meets the standards of clarity, conciseness, and academic rigor.
1. Manuscript length and repetition
“The manuscript is overly long, especially for a single case report. Many sections, particularly the case presentation and discussion, are excessively wordy and sometimes repetitive. Focus on the most relevant clinical, surgical, and technological points. Consolidate redundant information.”
Response:
Thank you for this important observation. We have undertaken a comprehensive revision of the manuscript, especially the Case Presentation
2. Grammar, language clarity, and tone
“Numerous grammatical errors, convoluted sentence structures, and awkward phrasing detract from both readability and scientific professionalism… Repetitive phrases such as ‘this case illustrates...’ and ‘this case demonstrates...’ occur too frequently.”
Response:
We greatly appreciate your emphasis on academic clarity. The entire manuscript has been carefully reviewed and revised to address all grammatical inconsistencies, eliminate awkward or overly colloquial phrasing, and replace repetitive sentence structures. Phrases such as “this case demonstrates...” have been varied or removed where appropriate, and technical terms have been tightened to reflect standard scientific language.
3. Narrative structure and flow
“The narrative is sometimes hard to follow due to mixing clinical chronology with broader conceptual digressions. Consider reorganizing sections – especially 'Case presentation' – by adding subsections like 'surgical technique' and 'postoperative course', etc.”
Response:
Thank you for this insightful suggestion, which we considered with great care. We fully agree that structuring the Case Presentation into clear subsections such as Surgical Technique and Postoperative Course would enhance clarity and narrative coherence. However, we have chosen to preserve the unified format in alignment with the journal’s stylistic preferences for case reports, which generally emphasize continuous, integrated narration over segmented subheadings.
4. Anatomical terminology and accuracy
“Occasional inconsistency in anatomical terminology, e.g., ‘distal subclavian’ likely meant ‘distal PICA’, and some vague or inaccurate references, e.g., ‘retro vertebral and thoracic anatomy’. Ensure anatomical terms are accurate and standard.”
Response:
We fully agree and thank you for identifying this issue. The anatomical terminology throughout the manuscript has been reviewed and corrected for consistency and precision.
5. AI, genetics, and computational modeling content
“While insightful, the integration of AI, genetics, and computational models is too broad and speculative for a single case. Keep such content concise, limiting to how it specifically relates to this case.”
Response:
Thank you for raising this important concern.
6. Citation accuracy and completeness
“Double check citations. Some parts are lacking references, e.g., lines 48–51, 54–56, 61–63, 65–67, 87–92, 453–456, 461–464, 470–472, 475–480, 485–487, 491–493, 509–511, 525–529, 532–536.”
Response:
We thank you sincerely for identifying these gaps. The manuscript has been reviewed line by line, and all missing references have now been inserted. In total, we added several key citations to support claims related to aneurysm morphology, genetics, microsurgical strategy, and emerging technologies. References were double-checked for formatting consistency and completeness across the manuscript.
We are deeply appreciative of the time and care you devoted to this review. Your feedback has been instrumental in improving the manuscript’s clarity, precision, and academic value. We hope the revised version now meets the publication standards of the journal and respectfully submit it for your reconsideration.
With sincere thanks and professional appreciation!
Round 2
Reviewer 2 Report
Comments and Suggestions for Authors
The article has been improved. All the changes based on our suggestions have been addressed.
Author Response
Dear Academic Reviewer,
We sincerely thank you for your kind and encouraging feedback. We are truly grateful that you found the revised version of our manuscript improved and that the changes we implemented in response to your thoughtful suggestions have met your expectations.
Your constructive insights were invaluable in refining the structure, clarity, and scientific rigor of the manuscript, and we deeply appreciate the time and care you devoted to reviewing our work. It was an honor to incorporate your recommendations, and we are hopeful that the final version now presents a clearer and more cohesive contribution to the field.
Thank you once again for your generous support and for helping us strengthen the quality of our submission.
With highest respect and appreciation,
The Authors